# Enhancing Journalism with AI: A Study of Contextualized Image Captioning for News Articles using LLMs and LMMs

**Aliki Anagnostopoulou**[1,2] , **Thiago S. Gouvêa**[1] and **Daniel Sonntag**[1,2]

[1] DFKI German Research Center for Artificial Intelligence
[2] Applied Artificial Intelligence, Carl von Ossietzky University Oldenburg
{firstname.lastname}@dfki.de,

## Abstract

Large language models (LLMs) and large multi-modal models (LMMs) have significantly impacted the AI community, industry, and various economic sectors. In journalism, integrating AI poses unique challenges and opportunities, particularly in enhancing the quality and efficiency of news reporting. This study explores how LLMs and LMMs can assist journalistic practice by generating contextualised captions for images accompanying news articles. We conducted experiments using the Good-News dataset to evaluate the ability of LMMs (BLIP-2, GPT-4v, or LLaVA) to incorporate one of two types of context: entire news articles, or extracted named entities. In addition, we compared their performance to a two-stage pipeline composed of a captioning model (BLIP-2, OFA, or ViT-GPT2) with post-hoc contextualisation with LLMs (GPT-4 or LLaMA). We assess a diversity of models, and we find that while the choice of contextualisation model is a significant factor for the two-stage pipelines, this is not the case in the LMMs, where smaller, open-source models perform well compared to proprietary, GPT-powered ones. Additionally, we found that controlling the amount of provided context enhances performance. These results highlight the limitations of a fully automated approach and underscore the necessity for an interactive, human-in-the-loop strategy.

## 1 Introduction

Large language pre-training [Devlin *et al.*, 2019; Liu *et al.*, 2023c] and large vision-language pre-training [Wang *et al.*, 2022; Zou *et al.*, 2023], facilitated by advances in deep learning and the development of the Transformer [Vaswani *et al.*, 2017] architecture, have significantly impacted research and industry. In journalism, these models offer potential for human-AI collaboration; however, generating news articles with these models is not feasible since these pre-trained models lack up-to-date information on current events [Bubeck *et al.*, 2023], among other reasons. Instead, they can assist journalists by automating specific tasks, such as captioning images that accompany existing news articles. These captions

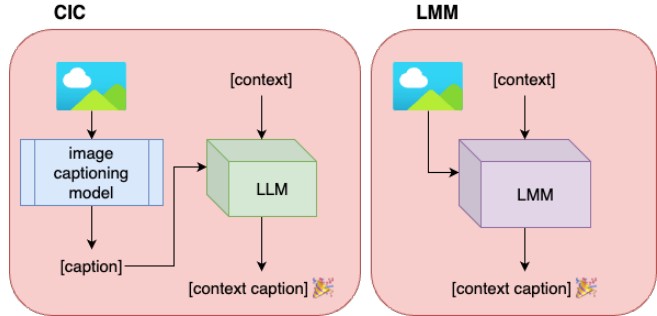

Figure 1: Our proposed architectures. Our two-stage pipeline CIC involves an image captioning system; the generated caption and contextual information are fed into an LLM. We compare this architecture with LLMs, which consider visual and textual input, omitting the need for an additional image captioning component.

should describe the image content *and* provide relevant contextual information that cannot be deduced from the image, including the names of people, locations, and events.

In this study, we investigate the effectiveness of large foundation models, specifically large language models (LLMs) and large multimodal models for vision-language tasks (LMMs) in performing *contextualised image captioning*. We conduct experiments using the GoodNews dataset, which contains contextualised image captions from news articles. We propose a two-stage pipeline composed of an image captioning model with post-hoc contextualisation performed by an LLM. We compare this pipeline, which we denote as CIC, with LLMs (see Figure 1). We evaluate nine configurations, including open-source models such as Llama 3 and LLaVA [Liu *et al.*, 2023b; Liu *et al.*, 2023a; Liu *et al.*, 2024] and closed-source models such as GPT-3 [Liu *et al.*, 2023c] and GPT-4v [OpenAI, 2023].

After briefly presenting related work to AI in journalism and contextualised image captioning (Section 2), we describe our pipelines, the foundation models used, the dataset, and the evaluation metrics in Section 3. We then present and discuss our results in Section 4. Section 5 concludes our work and discusses possible future directions.

| Image | Caption | Article | NE |
|---|---|---|---|
| 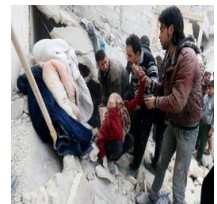 | Residents and activists aided a girl who survived amid debris in **Aleppo** on **Sunday** after what activists said was an aerial attack that dropped explosive barrels. | ISTANBUL – Syrian government aircraft continued to strike rebel-held areas in Aleppo with makeshift bombs on Sunday, killing at least three dozen people, most of them women and children, antigovernment activists said. [...] The government has not commented on the airstrikes other than to mention in the state news media that its forces have killed "terrorists," a blanket term for the opposition. | GPE: Aleppo; DATE: Sunday |

Table 1: Example image, caption and relevant context (article and extracted NEs) from the GoodNews dataset. The extracted NEs are also marked in the caption.

## 2 Related Work

This section reviews previous research relevant to our study, focusing on the application of AI in journalism and the field of contextualised captioning.

**LLMs and LMMs in journalism** [del Barrio and Gática-Pérez, 2023] use GPT-3.5 for news frame classification using fine-tuning and prompt engineering. [Bao *et al.*, 2024] develop a model specialised for question answering and data visualisation in the business and media domain. Following a human-centric approach, [Cheng *et al.*, 2024] propose a model incorporating human input for generating sports news insights.

As mentioned in Section 1, LLMs are unsuitable for generating news articles due to multiple shortcomings. Besides not being up-to-date, making them prone to hallucinations, LLM news generations can be biased: [Fang *et al.*, 2023] evaluate the gender and racial biases reproduced in LLM-generated content. [Hamilton and Piper, 2022] use GPT-2 to generate counterfactual news articles, finding out that they exhibit a notably more negative attitude towards COVID and a significantly reduced reliance on geopolitical framing.

**Contextualised image captioning** Contextualised image captioning considers additional context to generate an image caption that describes the image's content and includes relevant external information. The context provided is, in most cases, in textual form. [Biten *et al.*, 2019] and [Tran *et al.*, 2020] use news articles as context; the former uses a template-based architecture, and the latter uses an end-to-end architecture, considering additional features such as face and object detection. A modified version of the latter is used in [Nguyen *et al.*, 2023] for image captioning on Wikipedia [Srinivasan *et al.*, 2021], while an additional face naming module is also present in [Qu *et al.*, 2023]. [Rajakumar Kalarani *et al.*, 2023] present a unified architecture for context-assisted image captioning, including contextual visual entailment and keyword extraction.

## 3 Methods

This section describes our contextualising captioning experiments, namely the pipelines, type of context used, and evaluation (including datasets and metrics).

**TEXT PROMPT - GOODNEWS:**

You are a journalist. Describe this caption in a single sentence so the description suits a news article: `[image caption]`. Take the following context information into consideration: `[context info]`

You are a journalist. Describe this image in one sentence so the description suits a news article. Take the following context information into consideration: `[context info]`

Table 2: Proposed prompt for generating contextualised image captions using the GoodNews news dataset. In the upper row, the prompt for the CIC pipeline is present, and in the lower row the one for LMM.

### 3.1 Pipelines

We use two approaches to contextualise captions, which we describe below. In the first one, we pair a conventional image captioning (**CIC**) architecture with an LLM for post-hoc contextualisation. In the second one, we utilise large multimodal models (**LMM**s), in which the image is directly provided as input, along with the context. Both pipelines are presented in Figure 1.

**CIC** As mentioned above, this approach follows two stages. In the base captioning stage, the image captioning architecture generates a description for a given image. In the contextualising stage, the LLM takes the generated caption and the context as input and generates a caption that includes the context for each image. Such an approach might be beneficial if access to LLMs is not possible, if privacy issues are present, or if a base caption is needed a priori. A drawback in this case, however, is the *information bottleneck* caused by the contextualising model not having access to all the visual information in the image. We use pre-trained SOTA models for base image captioning: ViT-GPT2[1], OFA [Wang *et al.*, 2022], and BLIP-2 [Li *et al.*, 2023]. For contextualisation, we use GPT-3(.5) [Liu *et al.*, 2023c] and Llama 3[2].

**LMM** LMMs can process visual and text input simultaneously; hence, there is no explicit intermediate caption generation process in this case. BLIP-2, used in the CIC configu-

---

[1]https://huggingface.co/nlpconnect/vit-gpt2-image-captioning
[2]https://llama.meta.com/llama3/

| | | | BLEU | | | | ROUGE | | | METEOR | BERTScore | | | SBERT |
|---|---|---|---|---|---|---|---|---|---|---|---|---|---|---|
| | | | 1 | 2 | 3 | 4 | 1 | 2 | L | | p | r | f1 | |
| CIC | GPT-3 | BLIP-2 | **0.373** | **0.182** | **0.101** | **0.060** | **0.356** | **0.184** | **0.292** | **0.344** | **0.897** | **0.900** | **0.898** | **0.526** |
| | | OFA | 0.368 | 0.178 | 0.097 | 0.057 | 0.352 | 0.182 | 0.289 | 0.341 | 0.895 | 0.898 | 0.896 | 0.507 |
| | | ViT-GPT2 | 0.363 | 0.175 | 0.096 | 0.057 | 0.345 | 0.177 | 0.283 | 0.334 | 0.894 | 0.897 | 0.895 | 0.482 |
| | Llama | BLIP-2 | 0.051 | 0.020 | 0.009 | 0.005 | 0.119 | **0.051** | **0.103** | 0.211 | 0.742 | 0.860 | 0.796 | **0.632** |
| | | OFA | 0.052 | 0.020 | 0.010 | 0.005 | 0.119 | 0.050 | **0.103** | 0.211 | 0.745 | 0.860 | 0.797 | 0.630 |
| | | ViT-GPT2 | **0.062** | **0.024** | **0.012** | **0.006** | **0.123** | **0.051** | 0.101 | **0.220** | **0.788** | **0.865** | **0.824** | 0.586 |
| LMM | | BLIP-2 | **0.336** | **0.137** | 0.062 | 0.036 | 0.304 | 0.143 | 0.266 | 0.194 | 0.882 | 0.856 | 0.868 | 0.402 |
| | | GPT-4v | 0.283 | 0.114 | 0.061 | 0.035 | **0.328** | **0.151** | **0.248** | **0.322** | 0.872 | **0.888** | **0.879** | **0.505** |
| | | LLaVA | 0.331 | 0.132 | **0.072** | **0.043** | 0.283 | 0.129 | 0.246 | 0.260 | **0.885** | 0.872 | 0.878 | 0.399 |

Table 3: Similarity between ground truth captions (GoodNews dataset) and those generated with named entity context. BERTScore: micro-averaged precision (p), recall (r), and F1 score.

ration, can also take textual instructions as input, functioning as an LMM. We additionally consider two additional LMMs, namely GPT-4v and LLaVA. All GPT models are provided by OpenAI, while the others are open-source and publicly available on the Huggingface platform[3].

**Prompting** Since LLMs and LMMs have different input requirements, we use two prompt versions, slightly modified. The prompts are present in Table 2. The CIC pipeline must include the base image caption and context to generate the appropriate contextualised caption. In contrast, for the LMM pipeline, only the context must be explicitly provided in the text prompt.

## 3.2 Evaluation

**Dataset** We use a subset of the GoodNews dataset [Biten *et al.*, 2019]. This dataset contains images and articles, along with contextualised captions. An example is provided in Table 1. We only consider images from 1,000 articles, resulting in a total 1,791 images with their respective captions.

**Metrics** To assess the quality of the generated captions, we use natural language generation metrics: BLEU [Papineni *et al.*, 2002], ROUGE [Lin, 2004], METEOR [Banerjee and Lavie, 2005], and BERTScore [Zhang *et al.*, 2020]. The first three methods are older and widely used to evaluate natural language generation tasks, such as machine translation and image captioning, relying on n-gram overlaps between reference and generated text. BERTScore, on the other hand, was introduced more recently. It leverages the pre-trained contextual embeddings from BERT [Devlin *et al.*, 2019] and matches words between reference and generated text by cosine similarity. We additionally measure sentence embedding similarity with a pre-trained SBERT [Reimers and Gurevych, 2019] model.

## 3.3 Context types

As seen in Table 5, we consider two kinds of textual context. In the first case, we extract relevant *named entities* (NE), from the target captions, such as person or organisation names, locations, and time. This way, contextualisation can focus on

---

[3]https://huggingface.co/

the information appropriate to the caption. In the second case, we consider the whole *article* as context. This provides a larger amount of information to the systems, which, in turn, might not be relevant. Technically speaking, in a larger application, providing the article as context would equal a less controllable but less labour-intensive approach than providing extracted explicit entities.

## 4 Results and discussion

Table 3 and Table 4 show the results for generating contextualised captions given NE context and article context, respectively. In CIC models, we observe a significantly lower performance when using Llama 3. It is interesting, however, that there is no significant difference between CIC with GPT-3 and LMM. In this case, the bottleneck caused by the lack of visual information beyond the text caption is not substantial - probably because context information contributes more to the meaning of the caption than the content. This is particularly interesting in the comparison between GPT-4v and LLaVA: BLEU-3 and -4 scores are higher for LLaVA, and BERTScores between the two models do not differ significantly. This indicates that similar results can be achieved both with closed- and open-source models.

**Focused context matters** Results in Table 3 are in almost all cases significantly higher than in their respective categories and metrics in Table 4 - in the case of BLEU-3 and -4, which measures trigram and tetragram overlap, scores are close to or equal to zero. This indicates that less is more; since a model is prone to hallucinating and not correctly following the instruction in the text prompt, a more controlled approach where only needed information must be explicitly mentioned in the caption is more beneficial. However, this might cause more overhead for the domain expert, as the relevant context must be identified and integrated manually. In this case, an approach facilitating these processes by interaction and/or integration of additional, controllable modules would prove beneficial.

## 4.1 Ablation study

As an additional ablation experiment, we calculate BERTScore values between our reference ground truth

| | | | Metrics | | | | | | | | | | | |
|---|---|---|---|---|---|---|---|---|---|---|---|---|---|---|
| | | | BLEU | | | | ROUGE | | | METEOR | BERTScore | | | SBERT |
| | | | 1 | 2 | 3 | 4 | 1 | 2 | L | | p | r | f1 | |
| CIC | GPT-3 | BLIP-2 | **0.188** | **0.034** | **0.013** | **0.006** | **0.171** | **0.044** | **0.136** | **0.164** | **0.858** | **0.855** | **0.857** | **0.323** |
| | | OFA | 0.114 | 0.002 | 0.000 | 0.000 | 0.071 | 0.002 | 0.059 | 0.074 | 0.831 | 0.824 | 0.828 | 0.314 |
| | | ViT-GPT2 | 0.116 | 0.002 | 0.000 | 0.000 | 0.072 | 0.002 | 0.059 | 0.075 | 0.831 | 0.825 | 0.828 | 0.304 |
| | Llama | BLIP-2 | **0.046** | **0.010** | **0.004** | **0.002** | **0.084** | **0.021** | **0.067** | **0.152** | **0.788** | **0.844** | **0.815** | **0.567** |
| | | OFA | 0.033 | 0.002 | 0.000 | 0.000 | 0.049 | 0.003 | 0.042 | 0.089 | 0.779 | 0.815 | 0.796 | 0.563 |
| | | ViT-GPT2 | 0.032 | 0.002 | 0.000 | 0.000 | 0.049 | 0.003 | 0.042 | 0.089 | 0.778 | 0.815 | 0.796 | 0.561 |
| LMM | | BLIP-2 | **0.195** | **0.044** | **0.018** | **0.009** | **0.166** | **0.046** | **0.142** | **0.102** | **0.853** | **0.838** | **0.845** | 0.270 |
| | | GPT-4v | 0.105 | 0.003 | 0.000 | 0.000 | 0.092 | 0.003 | 0.074 | 0.097 | 0.82 | 0.824 | 0.822 | **0.366** |
| | | LLaVA | 0.131 | 0.004 | 0.000 | 0.000 | 0.094 | 0.004 | 0.08 | 0.078 | 0.834 | 0.824 | 0.829 | 0.276 |

Table 4: Similarity between ground truth captions (GoodNews dataset) and those generated with article context. BERTScore: micro-averaged precision (p), recall (r), and F1 score.

| | | NE | | article | |
|---|---|---|---|---|---|
| | base | GPT | Llama 3 | GPT | Llama 3 |
| BLIP-2 | 0.841 | 0.891 ↑ | 0.796 ↓ | 0.857 ↑ | 0.815 ↓ |
| OFA | 0.840 | 0.889 ↑ | 0.797 ↓ | 0.828 ↓ | 0.796 ↓ |
| ViT-GPT2 | 0.854 | 0.887 ↑ | 0.824 ↓ | 0.828 ↓ | 0.796 ↓ |

Table 5: Ablation study: F1 BERTScores for base captions, compared to contextualised ones (given different contexts and post-hoc contextualisation LLMs).

captions and base captions generated by our three image captioning models of choice. Results for experiments with both contexts are present in Table 5. We expect that the base captions would perform worse than their contextualised counterparts. However, this is only the case with GPT-3 in combination NE context - compared to the base, non-contextualised captions, scores for the contextualised captions with Llama 3 is lower. This indicates Llama 3's inability to follow the prompt's instructions as expected, which leads to caption generations containing irrelevant information, lowering the automated metric scores.

### 4.2 Limitations

We identify two significant limitations within our work. The first is the *lack of diversity in prompt usage*. We experiment with a single prompt as present in Table 2. Especially in cases like CIC with a Llama 3 contextualisation module, experimentation with prompts might be beneficial and lead to improved results. The second limitation is related to the *metrics* we use. The "older" methods (BLEU, ROUGE, METEOR) might not reward the existence of synonyms and paraphrases, as they focus on exact matches and their order. A domain expert, in this case, a journalist, might consider one of the penalised captions just as well-formed as its ground truth equivalent. On the other hand, BERTScore might be rather forgiving - hence the higher scores in the tables. When accuracy is required, however, the scores might not capture the difference in generated quality. Thus, a user study is necessary to evaluate the presented pipelines' performance.

## 5 Conclusion and future work

The appearance of LLMs and LLMs has rendered human-AI synergy more accessible. This work presents a use case for journalism: generating relevant, contextualised image captions given different pipelines (called CIC and LMM), including pre-trained image captioning models, LLMs, and LMMs. We evaluate our experiments with automated metrics and conclude that, at least regarding these metrics, the bottleneck caused by using a CIC-like architecture with a textual description of the image rather than the image itself is insignificant. Close-source models such as the GPT family might have an advantage in the CIC configuration. However, smaller, open-source models perform similarly well in the LLM configuration.

In terms of context, focused information, such as NEs, is more beneficial to the models than the whole article itself. This finding indicated a possible future direction for our work: implementing an interactive system that facilitates journalists' writing captions for their articles. Additionally, we would like to expand our contextualised image captioning experiments to include more datasets, and address the limitations stated in Section 4, by experimenting with different prompt patterns to increase the efficiency of the proposed architectures and by conducting a user study for a more nuanced evaluation of the quality of the generated captions.

## Ethical Statement

We have carefully considered the ethical implications of our work and do not foresee any major concerns. The dataset and models we utilize are publicly available. However, we acknowledge the potential risks of disseminating incorrect information, which could lead to the misuse of LLMs and LMMs. This is an important limitation of our research that we strive to mitigate through rigorous evaluation and responsible usage guidelines.

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
