# OpenReview forum: "Enhancing Journalism with AI: A Study of Contextualized Image Captioning for News Articles using LLMs and LMMs"
_ijcai.org/IJCAI/2024/Workshop/TIDMwFM — IJCAI TIDMwFM 2024 Poster_

### Official Review · Reviewer_qixV · 2024-07-24
**Paper Review**

**Rating:** 7
**Confidence:** 3

**Review:**

Summary
In this paper, the authors investigate the application of LLMs andLMMs to enhance journalistic practices through contextualized image captioning for news articles. The study explores two main approaches: CIC that combines image captioning models with post-hoc contextualization using LLMs, and direct use of LMMs that simultaneously process visual and textual inputs. The experiments, conducted on the GoodNews dataset, evaluate the performance of various models, including open-source llama, LLaVA and GPT-4v. The results show that smaller, open-source LMMs perform comparable to proprietary models, and focused context is more beneficial than whole articles for generating relevant captions.

Strengths and Novelty
This paper's novelty lies in the comparative analysis of different models and configurations, highlighting the potential of open-source models to perform comparable to with proprietary ones. The use of metrics to evaluate caption qualityis also very reasonable. The study's exploration of different context types (named entities vs. full articles) provides valuable insights into optimizing AI-assisted journalism workflows. Overall, a very novel work whose insights can be leveraged for using foundation models in the journalism, and it highly aligns with the theme of the workshop.

Areas of Improvement and Feedback
While the paper presents novel and interesting findings, some other things can also be included. First, exploring a variety of prompts could enhance the robustness of the results. Second, the paper does not fully address the overhead associated with manually identifying and integrating relevant context, suggesting a need for automated or semi-automated solutions.

---

### Decision · Program_Chairs · 2024-07-24

**Decision:**

Accept (Poster)

**Comment:**

This paper is highly relevant to the theme of the workshop and offers valuable insights of using the foundation models in the journalism. I recommend the paper to be accepted to the workshop.